The hypoxia-associated genes in immune infiltration and treatment options of lung adenocarcinoma

Liu Liu 1
Han Lina 1
Dong Lei 1
He Zihao 1
Gao Kai 1
Chen Xu 1
Guo Jin-Cheng guojincheng@bucm.edu.cn 1
Zhao Yi zhaoyi@bioinfo.org 1 2
1 School of Traditional Chinese Medicine, Beijing University of Chinese Medicine , Beijing , China
2 The Research Center for Ubiquitous Computing Systems (CUbiCS), Institute of Computing Technology, Chinese Academy of Sciences , Beijing , China
Tyagi Abhishek
Electronic publication date: 2023 Aug 7
Publication date: 2023
Volume: 11
Electronic Location ID: e15621
Received 2023 Mar 23; Accepted 2023 Jun 1
Copyright: ©2023 Liu et al.
Copyright year: 2023
Copyright holder: Liu et al.
License: This is an open access article distributed under the terms of the Creative Commons Attribution License, which permits unrestricted use, distribution, reproduction and adaptation in any medium and for any purpose provided that it is properly attributed. For attribution, the original author(s), title, publication source (PeerJ) and either DOI or URL of the article must be cited.
License URL: https://creativecommons.org/licenses/by/4.0/

Keywords: Hypoxia, Lung adenocarcinoma, Unsupervised clustering, Prognostic, Immunotherapy

Funding: National Natural Science Foundation of China 32000439 This work was supported by the National Natural Science Foundation of China (32000439). The funders had no role in study design, data collection and analysis, decision to publish, or preparation of the manuscript.

==============================
Background

Lung adenocarcinoma (LUAD) is a common lung cancer with a poor prognosis under standard chemotherapy. Hypoxia is a crucial factor in the development of solid tumors, and hypoxia-related genes (HRGs) are closely associated with the proliferation of LUAD cells.

Methods

In this study, LUAD HRGs were screened, and bioinformatics analysis and experimental validation were conducted. The Cancer Genome Atlas (TCGA) and the Gene Expression Omnibus (GEO) databases were used to gather LUAD RNA-seq data and accompanying clinical information. LUAD subtypes were identified by unsupervised cluster analysis, and immune infiltration analysis of subtypes was conducted by GSVA and ssGSEA. Cox regression and LASSO regression analyses were used to obtain prognosis-related HRGs. Prognostic analysis was used to evaluate HRGs. Differences in enrichment pathways and immunotherapy were observed between risk groups based on GSEA and the TIDE method. Finally, RT-PCR and in vitro experiments were used to confirm prognosis-related HRG expression in LUAD cells.

Results

Two hypoxia-associated subtypes of LUAD were distinguished, demonstrating significant differences in prognostic analysis and immunological characteristics between subtypes. A prognostic model based on six HRGs (HK1, PDK3, PFKL, SLC2A1, STC1, and XPNPEP1) was developed for LUAD. HK1, SLC2A1, STC1, and XPNPEP1 were found to be risk factors for LUAD. PDK3 and PFKL were protective factors in LUAD patients.

Conclusion

This study demonstrates the effect of hypoxia-associated genes on immune infiltration in LUAD and provides options for immunotherapy and therapeutic strategies in LUAD.

Introduction

Non-small cell lung cancer (NSCLC) is one of the most common malignancies worldwide and consists of two main types, lung adenocarcinoma (LUAD) and lung squamous cell carcinoma (LUSC), with LUAD having a higher incidence (Graham & Unger, 2018). In the current clinical diagnosis of NSCLC, more than 70% of NSCLC patients only start treatment in the late stage of cancer due to missing the early treatment stage (Tsubata, Tanino & Isobe, 2021), which leads to the ineffectiveness of conventional chemotherapy. In recent years, with the emergence of molecularly targeted therapy technologies, various new immunotherapies have been administered for LUAD (Herbst, Morgensztern & Boshoff, 2018; Yang et al., 2020; Shao et al., 2021; Sun et al., 2020), and the prognosis of LUAD patients has significantly improved (Tsubata, Tanino & Isobe, 2021). However, issues such as immunotherapy regimens and efficacy prediction markers still need extensive exploration. As a result, more research on novel possible prognostic biomarkers is required to effectively assess the prognosis of tumor patients and provide a basis for individualized LUAD diagnosis and treatment.

Hypoxia is essential in the microenvironment of solid tumors (Graham & Unger, 2018; Wigerup, Pahlman & Bexell, 2016). Solid tumor tissues are usually poorly oxygenated and lack functional vasculature, resulting in a chronic hypoxic state of the tumor microenvironment (Ruan, Song & Ouyang, 2009). Hypoxia affects various functions such as cell metabolism and cell apoptosis, and promotes tumor growth (Lu & Kang, 2010; Harris, 2002). As a major factor in cell adaptation to hypoxia (Cowman & Koh, 2021), hypoxia-inducible factor (HIF) can alter the malignant potential of the tumor and promote the growth of associated blood vessels.

The impact of hypoxic factors on the microenvironment of LUAD tumors remains to be investigated. Methods to identify patients at high risk of LUAD based on hypoxia-related factors and to predict prognosis still need improvement. Numerous studies have investigated the impact of hypoxia-related genes on immune infiltration and prognosis in LUAD cancer. The classical approach involves constructing a prognostic model using hypoxia-related genes as variables to evaluate their influence on LUAD prognosis. However, this method cannot directly assess the impact of gene expression on immune infiltration and patient survival, as prognostic information is analyzed as a variable. Alternatively, many current cancer studies use clustering methods to divide clinical cohorts into subgroups based on specific types of gene expression, which effectively evaluate the impact of specific gene sets on immunity and prognosis while eliminating the interference of prognostic data and other information. Therefore, this study use clustering algorithms to distinguish clinical subtypes based on hypoxia gene expression, and then evaluates the impact of hypoxia-related genes on the immune microenvironment of LUAD tumors using immune analysis tools. Furthermore, we assess the clinical prognostic impact of hypoxia-related genes on lung adenocarcinoma patients using prognostic analysis tools and validate our findings using in vitro experiments. This study aims to analyze the effects of hypoxia-related genes on immune infiltration in LUAD, identify prognosis-related HRGs, construct a reliable prognostic profile for LUAD, and explore potential hypoxia-based therapeutic options in LUAD. We aim to identify effective hypoxia-related prognostic biomarkers that could provide support for clinical treatment.

Materials & Methods

Data acquisition of LUAD patients

The analysis process of this study is shown in the flow chart (Fig. 1). The UCSC Xena Browser (https://xenabrowser.net/) was used to download mRNA expression and clinical information for LUAD patients in TCGA, with 503 tumor samples and 35 normal samples. The samples of “primary solid tumors” were selected for follow-up studies. The downloaded RNA-seq dataset was in FPKM format, which reflected the actual expression quantity of the gene when comparing multiple samples. RNA-seq data and clinical information, including 695 tumor samples from three LUAD cohorts from the Gene Expression Omnibus (GEO) (GSE72094, GSE31210, GSE30219) were acquired. To ensure the reliability of the study, with an overall survival (OS) time of ≥30 days were retained and unexpressed genes in ≥50% of the samples were removed to improve data quality. A total of 1198 LUAD patients from four datasets were included. Detailed clinical information for LUAD patient clinical cohorts is summarized in Table S1.

Figure 1 Research workflow.

Identification of survival-related HRGs in LUAD

HRGs were acquired from the MsigDB database (https://www.gsea-msigdb.org/gsea/msigdb), and the latest HRGs were saved in an ensemble format. Univariate Cox model regression analysis was performed to screen further the GSE72094 cohort for HRGs associated with patient OS information (P value <0.05).

Identification of hypoxia subtypes of the LUAD cohort based on consensus clustering

As a typical unsupervised clustering method, consensus clustering is a common method for classifying cancer subtypes (Lock & Dunson, 2013). The prognosis-related HRG expression screened by univariate Cox analysis was used to subtype the LUAD cohort in TCGA and three GEO databases, and the most reasonable clustering results were determined by comparison.

Immunological characteristics and clinical features of hypoxia subtypes

Based on human genetic set data, samples from TCGA were subjected to GSVA analysis to obtain pathway scores, and formal difference analysis of pathways between hypoxic subtypes was performed using the limma package (Ritchie et al., 2015) to assess differences in subtype immunological characteristics (Hanzelmann, Castelo & Guinney, 2013). Single-sample gene set enrichment analysis (ssGSEA) was based on 29 immune cell types to quantify immune infiltration in the TCGA cohort using the “GSVA” package (Zhao, Ou & Lin, 2021). Differences in clinical characteristics between the hypoxic subtypes were also evaluated.

HRG prognostic signature development and validation

The GSE72094 dataset was used as a training group based on careful consideration of the number of clinical patients and the sequencing platform. It was tested in three other independent datasets (i.e., TCGA-LUAD, GSE31210 and GSE30219) for the predictive effect of these features to find potential prognostic biomarkers. HRGs related to survival in the training group were further selected and validated by LASSO regression using the “glmnet” package. Multivariate Cox model regression analysis was conducted to filter the HRGs associated with ultimate survival to develop a risk signature for LUAD patients. The characteristics with univariate Cox model results in P < 0.05 were selected for inclusion in the multivariate Cox model analysis. The risk score was calculated using the formula: risk score = ∑coefi * HRGi, where coefi represents the coefficient of i HRG in multivariate Cox model regression analysis, and HRGi denotes the expression level.

LUAD patients were divided into high- and low-risk groups based on the median value of the risk score. Kaplan–Meier (KM) survival analysis and time-ROC curves were used to evaluate the risk model performance in predicting prognosis.

Deterministic curve analysis (DCA) is a method to determine the clinical utility of different predictive models (Kerr et al., 2016). Calibration plots were used to verify the discrimination of the nomograms. The “index.comp” package compared the C-index between different groups.

Tumor mutation burden analysis

Based on somatic mutation data from TCGA, TMB information was obtained for each patient sample in the LUAD-TCGA cohort, and the difference in mutated genes between hypoxic subtypes or risk groups was compared by the R package “maftools”.

Tumor-infiltrating immune cells (TIICs) and immune checkpoint inhibitor analysis

We utilized the MCP-COUNTER (Aran, Hu & Butte, 2017) algorithm to compare immune infiltration differences among risk groups in LUAD-TCGA through the “immunedeconv” package. Additionally, we utilized the “estimate” package to calculate tumor purity scores based on the ratios of immune and stromal cells, ultimately obtaining immune scores, stromal scores, and microenvironment scores.

To evaluate the response to immune checkpoint inhibitors (ICI), we employed the Tumor Immune Dysfunction and Exclusion (TIDE) algorithm, which is capable of predicting the effectiveness of immune checkpoint suppression therapy.

Gene set enrichment analysis

GSEA was performed to reveal KEGG pathways in diverse risk groups of TCGA-LUAD (Subramanian et al., 2005). P < 0.05 and —NES—>1 represent statistical significance. Gene-Concept Network was performed to show complex connections between genes and enriched pathways for visual network presentation by “cnetplot” in R.

Evaluation of candidate targeted drugs

In order to evaluate the predictive capability of the small molecule drug susceptibility risk model, the half-maximal inhibitory concentration (IC50) scores among all candidate targeted drugs were estimated between two risk groups using the “oncoPredict” package. The Connectivity Map (CMap) database (https://clue.io/) was applied to access potential molecule drugs that were highly correlated with both risk groups.

Survival analysis of single HRGs

Prognostic analysis was performed separately for each gene constituting the HRG risk model using sample data from the GSE72094 training set. A minimum P value approach was used to assess the optimal survival cutoff, and patients were divided into high- and low-risk groups based on overall survival.

RT–PCR

Reverse-transcriptase PCR was performed on HRGs. The control gene was GAPDH. The total RNA of A549 cells was extracted by the TRIzol method. The ex-tracted RNA was reverse transcribed into cDNA, followed by PCR amplification. Samples were preamplified at 95 °C for 10 min followed by 40 amplification cycles for 2 s at 95 °F, 20 s at 60 °C, and 10 s at 70 °C. Subsequently, the product DNA was electrophoresed on a 2% agarose gel. The primers used for RT–PCR assays are listed in Table S2. The A549 cell line was obtained from Dr. Jing Liu at the National Center for Nanoscience and Technology, China (NCNST).

Cell transfection analysis

The day before transfection, cells were seeded into 6-well plates, which guar-anteed 70%–90% confluence of cells at the time of transfection. RNAiMAX was used with 20 µmol/L siRNA- or negative control-transfected cells. For one well of the 6-well plate, 4 µL of siRNA and 9 µL of RNAiMAX were dissolved in 125 µL of opti MEM medium, mixed at room temperature, left to stand for 10 min, and then added dropwise to the wells. After 48 h, the transfected cells were collected, and some cells were isolated for total RNA according to a previous method. The knock-down efficiency was measured by RT–PCR with Vazyme’s one-step kit, and the cell phenotypes of the remaining cells were measured, such as proliferation, apoptosis, migration, and invasion. RT–PCR was performed using GAPDH as the internal ref-erence gene, and each group of experiments had three repetitions. siRNA sequences are shown in Table S3.

Cell proliferation analysis

Cell proliferation was assessed by a cell counting kit-8 (CCK-8) Kit. The CCK-8 assay is commonly used to measure cell proliferation, which involves the use of the WST-8 reagent. This reagent comprises of 2-(2-methoxy-4-nitrophenyl)-3-(4-nitrophenyl)-5-(2,4-disulfophenyl)-2H-tetrazolium monosodium salt, which can be reduced by dehydrogenases present in the mitochondria of viable cells. Upon reduction, a water-soluble orange-colored formazan dye is formed. In cell experiments, the Optical Density (OD) value, which indicates the degree of light absorption by the sample, is measured using a spectrophotometer. This value is usually represented as the measured absorbance value. The OD value measured at a wavelength of 450nm in the microplate reader can be used as an indirect measure of cell quantity and proliferation. Briefly, 1 × 103 cells were seeded in each 96-well plate, and initial values were recorded after attachment to the wells and further incubated for 1, 2, 3, 4, and 5 days. Then, 100 µL of CCK-8 reagent was added to each well, and absorbance was detected at 450 nm. There were 6 repli-cates of each set of experiments.

Flow cytometric assay for apoptosis

Apoptosis detection was conducted using an Annexin V-PE/7-AAD Apoptosis Detection Kit. The siRNA-transfected A549 cell lines were cultured for 48 h. Cells harvested with 0.25% trypsin were washed twice more with cold PBS. Cells were stained with annexin v-PE and 7-AAD. The apoptosis rate was detected by flow cytometry. Each set of experiments was repeated three times.

Transwell assay

Cell invasion status was analyzed using 24-well Transwell chambers. After transfection, cells were inoculated into the upper chamber at a density of 1 × 10 ˆ5 cells/well using 200 µL of blank medium, which was precoated with Matrigel. The lower chamber was filled with 600 µL of RPMI-1640 containing 20% FBS. Cells were cultured at 37 °C for 24 h, fixed in paraformaldehyde for 15 min, and stained with 0.5% crystal violet for 15 min. Cells on the upper side of the mem-brane were removed with a clean cotton swab, and the number of invading cells was observed with a Nikon Eclipse Ti fluorescence microscope at 20×, and five randomly selected areas were counted under the microscope.

“Scissor” algorithm to identify subpopulations of TAMs associated with the LUAD hypoxic subtype phenotype in single-cell sequencing data

The “Scissor” algorithm presents an innovative approach to identify specific subpopulations from single-cell data that exhibit a high correlation with phenotypic characteristics. This approach is based on integrating phenotypic information obtained from large volumes of RNA-seq data, which allows for a more comprehensive understanding of the cellular heterogeneity within a population (Sun et al., 2022). The algorithm quantified the similarity between bulk and single-cell sequencing data and subsequently optimized the regression model with a sample phenotype correlation matrix to determine subtype classification of single cells. Single-cell sequencing data were obtained from 10 normal lung and 10 fresh lung adenocarcinoma tissue samples taken from clinical patients and processed for single-cell RNA sequencing (Bischoff et al., 2021). TAMs in the single-cell data based on marker genes were screened for subsequent subgroup classification.

Results

Construction of survival-related HRGs in LUAD

A total of 214 HRGs were obtained from the MsigDB database. A total of 197 HRGs for 18,499 genes were then known to be identified in the GSE72094 dataset (n = 386). In the training group (GSE72094), 197 HRGs were used as variables to construct prognostic features and build a risk prediction signature. Univariate COX model regression analysis was performed to find a significant association with OS in LUAD patients. There were 62 survival-related HRGs in the GSE72094 dataset.

Identification of hypoxic subtypes of LUAD cohorts

The four LUAD cohorts were grouped into hypoxic subtypes based on the ex-pression levels of 62 survival-related HRGs using the unsupervised consensus clus-tering method. The analysis yielded that all four LUAD cohorts showed the best dif-ferentiation of clustering groups with K = 2 (Fig. 2A, Figs. S1A, S1C and S1E), with 189 patients belonging to hypoxia-related Cluster 1 and 314 patients in hypox-ia-related Cluster 2 of LUAD. The prognostic analysis of the TCGA-LUAD cohort assessed the survival differences between the two hypoxic subtypes. Kaplan −Meier curves showed significantly higher OS in Cluster 1 than in Cluster 2 (log-rank test, P < 0.0001; Fig. 2B). There were also significant differences in clinical characteristics such as tumor stage, Child grade, and histologic grade between the two groups. Similarly, the other three LUAD cohorts from the GEO database also showed the same prognostic outcome, i.e., Cluster 1 had a significantly higher survival advantage than Cluster 2 (log-rank test, P < 0.0001; Figs. S1B, S1D and S1F). The clustering of hypoxic subtypes based on the “Rtsne” package was subjected to t-SNE dimensionality reduction analysis to further verify that the two groups of hy-poxic subtypes identified based on Consensus Clusterin could be well differentiated (Fig. 2C). Subsequently, a Sankey diagram was created based on two hypoxia-related subtypes of LUAD to assess the distribution of clinical information in the hypoxic subgroups of the TCGA cohort. The proportion of patients with pathological stage 1 and alive was higher in Cluster 1 than in Cluster 2 (Fig. 2D). Prognostic analysis and clinical information analysis based on clustering groupings showed that the expression of HRGs interfered with the development of LUAD tumors.

Figure 2 Identification of Hypoxic subtypes of LUAD cohorts.

(A) Consensus clustering of the TCGA-LUAD cohort using hypoxia-associated genes (K = 2). (B) The Kaplan–Meier survival curves for the hypoxic subtypes Cluster 1 and Cluster 2 in the TCGA-LUAD cohort. (C) T-SNE diagrams of the hypoxic subpopulation of TCGA-LUAD. (D) Sankey diagrams of hypoxic subgroups and clinical characteristics of patients based on the TCGA-LUAD cohort.

Immunological characteristics and clinical features of the hypoxic subtypes

Immune infiltration analysis and comparison of the two hypoxic subtypes in the TCGA cohort. GSVA was used to explore the functional heterogeneity of metabolic pathways between the two hypoxic subgroups in LUAD, showing that Cluster 1 was significantly enriched in immune and inflammatory pathways such as complement and coagulation cascades and systemic lupus erythematosus (Fig. 3A), indicating that the enrichment of biological pathways associated with immune acti-vation influencing patients in Cluster 1 had a better prognostic outcome. Boxplot results showed that Cluster 1 had significantly higher ESTIMATE scores, immune scores, and stromal scores than Cluster 2 (Mann–Whitney U test, P < 0.001, Fig. 3B (1-3)) in TCGA cohort. TIDE analysis showed that the TIDE score was significantly lower in Cluster 1 than in Cluster 2 (P = 0.0028; Fig. 3B(4)), indicating that hypoxic Cluster 1 was more receptive to the effects of immunothera-py than Cluster 2. However, TMB analysis between the two hypoxic subgroups showed a higher TMB score in Cluster 1 than Cluster 2 (P < 0.0001; Fig. 3B(5)), predicting a better prognosis for patients with Cluster 1 who received immunotherapy. These studies demonstrated the impact of subtype classification based on the definition of hypoxia on the immune microenvironment and immune cell composition resulting from LUAD.

Figure 3 Immunological characteristics and clinical features of the hypoxic subtype.

(A) Analysis of GSVA based on hypoxic subtypes. (B) ESTIMATE score (1), immune score (2), stromal score (3), TIDE score (4), TMB score (5) between hypoxic subtypes. (C) ssGSEA analysis to assess differences in the expression of 28 immune cells between hypoxic subtypes. (D) Heat map of clinical features and immune cell expression based on combined ssGSEA analysis of hypoxic subtypes. * p-value < 0.05, ** p-value < 0.01, *** p-value < 0.001.

ssGSEA was used to quantify the differences in the abundances of 28 tumor-infiltrating immune cell types between different hypoxic subgroups. The ex-pression of immune cells such as activated CD4 T cells and myeloid-derived sup-pressor cells was significantly different between the hypoxic Cluster 1 and Cluster 2 subtypes (Fig. 3C). Among them, activated CD4 T cells and central memory CD8 T cells were signifi-cantly enriched in Cluster 1, while mast cells were significantly enriched in Cluster 2. Furthermore, analysis of the clinical characteristics of the patients showed statistically significant differences between the hypoxic subtypes (Fig. 3D). It was shown that the two subtypes differentiated by hypoxic features differ significantly in immune microenvironment and immune cell composition, leading to differences in clinical characteristics.

Role of hypoxia-related genes in predicting the prognosis of lung adenocarcinoma

To further analyze the impact of hypoxia genes on clinical prognosis, prognos-tic models were constructed based on patient mRNA data and clinical information. Sixty-two HRGs were analyzed using LASSO regression, and six key features were obtained (Figs. 4A and 4B). A risk profile of survival-related HRGs consisting of six genes (HK1, PDK3, PFKL, SLC2A1, STC1, and XPNPEP1) was constructed by multivariate Cox model regression analysis (Fig. 4C). Univariate and multivariate Cox model regression analyses were conducted for the six genes that ultimately constituted the risk signature (Table 1). The signatures used to calculate the risk score were as follows: risk score=0.7262630*HK1+−0.6109597*PDK3+−0.7375822*PFKL+0.3149958*SLC2A1+0.2331100*STC1+0.5576957*XPNPEP1.

Figure 4 LASSO regression prognostic model in the training set and multivariate COX model regression analysis and subgroup analysis for six-HRG signature.

(A) LASSO regression model to find the best optimal λ. (B) The process of changing LASSO coefficients of HRGs. (C) Forest plots of the Multivariate COX model regression analysis for six-HRG signature. (D) Subgroup analysis for six-HRG signature. * p-value < 0.05, ** p-value < 0.01, *** p-value ¡ 0.001.

Table 1 Univariate COX model and multivariate COX model analysis for six survival-related HRGs.

Gene Symbol	Univariable Cox regression	Multivariable Cox regression	
	coef	HR	95%CI	P -value	coef	HR	95%CI	P -value	
HK1	0.000622	1.000622147	1.00–1.01	<0.001	0.726263	2.07	1.23–3.48	0.006	
PDK3	0.001258	1.001258784	0.99–1.00	0.026	−0.6109597	0.54	0.42–0.7	<0.001	
PFKL	0.004348	1.00435754	1.00–1.01	<0.001	−0.7375822	0.48	0.28–0.81	0.007	
SLC2A1	0.0009	1.000900419	1.00–1.01	<0.001	0.3149958	1.37	1.17–1.61	<0.001	
STC1	0.000433	1.000432811	0.99–1.00	0.019	0.2331100	1.26	1.08–1.48	0.004	
XPNPEP1	0.001664	1.001665121	1.00–1.01	0.048	0.5576957	1.75	1–3.05	0.049	

The regression coefficients for the four HRGs, namely, HK1, SLC2A1, STC1, and XPNPEP1, were all positive, which indicated that they are related to poor prognosis. In contrast, PDK3 and PFKL were opposite protective factors in the prognostic signature.

For the training set GSE72094, univariate Cox model regression analysis among the cohort subgroups was performed to assess further the ability of risk characteristics (Fig. 4D). The results showed that risk scores were significantly associated with survival information in most subgroups of GSE72094 patients.

Training and validation of the survival prediction ability of six HRG features in the dataset

According to the median risk score based on the six-HRG signature, the cases in the training dataset GSE72094 were classified into high-risk (n = 193) or low-risk (n = 193) groups. Figure 5 shows the prognostic characteristics of the cohort. The Kaplan–Meier analysis curve showed significantly lower OS in the high-risk group than in the low-risk group (log-rank test, P < 0.0001; Fig. 5A). According to the results of univariate and multivariate Cox model analyses in multi-ple datasets, only the HRG signature was a significant prognostic risk factor (univariate Cox model P value <0.05 & multivariate Cox model P value <0.05), whereas the HRG signature was an independent risk factor for survival prediction (Table S4). Then, a time-dependent ROC analysis was established to evaluate the prognostic ac-curacy of the six-HRG signature. In the GSE72094 dataset, the AUCs for 1-, 3-, and 5-year survival were 0.66, 0.72, and 0.62, respectively (Fig. 5B).

Figure 5 Survival prediction capability of the six-HRG signature in the training and validation datasets.

(A) The Kaplan–Meier survival curves of high- and low-risk groups in training dataset GSE72094. (B) The AUC of time-dependent ROC curves in GSE72094 dataset. (C) DCA of the diagnostic nomogram in the training datasets. (D) The calibration curve of 3-year survival in the training datasets. The Kaplan–Meier survival curves, AUC of time-dependent ROC curves, DCA, and the calibration curve of 3-year survival in validation dataset GSE31210 dataset (E, F, G, H), GSE30219 dataset (I, J, K, L), and TCGA-LUAD dataset (M, N, O, P).

Three datasets were used to validate the survival prediction capability of the six-HRG signature, including GSE30219, GSE31210, and TCGA-LUAD (Figs. 5E–5P). KM analysis was conducted in verification sets, confirming the distinct difference in survival rate between the risk groups (Figs. 5E, 5I and 5M). The results of time-dependent ROC for the three validation cohorts are presented in Figs. 5F, 5J and 5N.

DCA of HRG signatures for 3-year OS in four LUAD clinical cohorts was per-formed to determine the nomograms’ clinical significance. The risk of death resulted in a significant positive net benefit (Figs. 5C, 5G, 5K and 5O). The calibration curves all showed high agreement between the observed and pre-dicted results (Figs. 5D, 5H, 5L and 5P). The C-indices of the 4 cohorts are shown in Table S5.

The heatmap introduced the risk score of six-HRG expression, while the dot plot showed the survival status in the training dataset GSE72094. The results showed that the expression of HK1, SLC2A1, STC1, and XPNPEP1 was higher in death-prone patients with high risk scores (Fig. 6A).

Figure 6 Information of the risk scores in the training dataset GSE72094 and waterfall plot of genes in risk groups in TCGA-LUAD.

(A) Expression heatmap of the six-HRGs, a dot plot of risk scores and survival status of LUAD patients in the training dataset GSE72094. (B) Waterfall plot of top 10 genes in high-risk group in TCGA-LUAD. (C) Waterfall plot of top 10 genes in the low-risk group in TCGA-LUAD.

Associations of the risk model with overall survival and clinicopathological characteristics of patients with LUAD

In order to investigate the relationship between the risk model and clinicopathological characteristics of LUAD patients, we conducted a Wilcoxon test on the differences in risk scores among subgroups categorized by age, gender, pathological stage, pathological T stage, pathological N stage, and pathological M stage in the TCGA-LUAD cohort. The results showed that the risk score was significantly correlated with age (P value <0.05), gender (P value <0.001), pathological stage (P value <0.05), pathological T stage (P value <0.01), and pathological N stage (P value <0.05), but not with pathological M stage (Fig. S2).

We further conducted stratified survival analysis on multiple LUAD subtypes categorized by age (<70 and ≥ 70 years), gender (Male and Female), tumor pathological stage (stage I–II and III-IV), pathological T stage (T1-2 and T3-4), pathological N stage (N0 and N1-3), and pathological M stage (M0 and M+), in order to assess the predictive performance of the six-HRG risk model for prognosis (Log-rank test, P value <0.05; Figs. S3A–S3F and S3H–S3L). Results demonstrated that patients in the high-risk group exhibited significantly poorer overall survival than those in the low-risk group across all LUAD subtypes, except for the pathological T1-2 subtype. These findings suggest that the six-HRG risk model is a reliable predictor for LUAD prognosis.

Analysis of tumor mutation burden in high- and low-risk groups in LUAD

In the LUAD-TCGA cohort, mutations in the TP53, TTN, MUC16, CSMD3, LRP1B, RYR2, USH2A, ZFHX4, and KRAS genes were present in the top 10 mutat-ed genes in both the high- and low-risk groups of clinical patients and exceeded 20% in the TCGA patient cohort (Figs. 6B and 6C). This indicates that the frequency of mutated genes is highly similar between the two risk groups.

Estimate of tumor-infiltrating immune cells and ICI analysis

The MCP-counter algorithm was used to assess the expression of various tumor-infiltrating immune cells (TIICs) in the LUAD-TCGA cohort and to compare the differential expression of TIICs between the high- and low-risk groups. The immune infiltration of tumor cells is associated with the prognosis of LUAD (Li et al., 2021). The Wilcoxon rank sum test showed that B cells, cancer-associated fibroblasts (CAFs), myeloid cells, and T cells were significantly different between the two risk groups. CAF infiltration was higher in the high-risk group. In contrast, B cells, myeloid cells, and T cells were more infiltrated in the low-risk group ( P value <0.05, Fig. 7A). Immunoinfiltration analysis was performed in three GEO datasets, including GSE72094 (Fig. S4A), GSE31210 (Fig. S4B), and GSE30219 (Fig. S4C), to evaluate the expression of immune cells in different risk groups. Results showed that CAFs, myeloid cells, and T cells exhibited statistical differences (P value <0.05) between risk groups in all three GEO datasets, and their expression patterns in risk groups were consistent with those in the TCGA-LUAD dataset. B cells showed significantly lower expression levels in the high-risk group compared to the low-risk group in GSE72094 and GSE31210 datasets (P value <0.05).

Figure 7 Estimate of tumor-infiltrating immune cells and ICI analysis, GSEA analysis and Evaluation of candidate targeted drugs.

(A) Estimates of tumor-infiltrating immune cells were calculated separately for the risk groups. Pearson’s correlation analysis between risk score and TMB score (B), stromal score (C), immune score (D), and microenvironment score (E), respectively, in the LUAD-TCGA cohort. (F) (1) Immune checkpoint inhibitors analysis of high-risk and low-risk groups. IC50 analysis of candidate small molecule drug Docetaxel (2), Erlotinib (3), Gefitinib (4), Paclitaxel (5), and Vinorelbine (6) in two risk groups in the LUAD-TCGA cohort. (G) KEGG pathway analyses between the high- and low-risk groups; ridge plot and Gene-Concept Network based on the GSEA analysis. * p-value < 0.05, ** p-value < 0.01, *** p-value < 0.001.

The xCell algorithm was used to analyze the immune score, stromal score, and tumor microenvironment score of the LUAD-TCGA cohort. Pearson’s correlation analysis was performed between the risk score and TMB, immune score, stromal score, and microenvironment score. A significant positive correlation between risk scores and TMB scores was shown (Fig. 7B). The results of the correlation analysis showed a statistically negative correlation between the risk score and all three immune infiltration scores (Figs. 7C–7E). The analyses suggest that the low-risk group exhibited higher immune activity, which may lead to a better prognosis and further validate the LUAD prognostic model based on HRGs in immunotherapy.

The TIDE score was used to analyze the effects of ICIs on the disease. The re-sults indicated that the high- and low-risk groups were less likely to evade tumor immunization by Wilcoxon test. Statistical analysis showed that the TIDE score was significantly lower in the high-risk group (P value <0.05), indicating that patients in the high-risk group may have a better therapeutic effect after receiving immunotherapy. (Fig. 7F(1)).

GSEA

To explore the potential mechanisms in different prognostic groups, GSEA between risk groups in LUAD-TCGA was performed. Based on the GSEA, 11 path-ways were significantly enriched (Table S6). “GPI-anchor biosynthesis” and “protein export” pathways were expressed in a more activated state in the high-risk group than in the low-risk group. In contrast, “Focal adhesion”, “Ribosome” and “Viral carcinogenesis” were expressed in a relatively inhibited state in the high-risk group. The top five pathways are shown in the ridgeplot and Gene-Concept Network (Fig. 7G).

Evaluation of candidate targeted drugs

Drugs approved for the treatment of NSCLC were collected from the DrugBank database (Sun et al., 2022) as targeted drugs. Using the Wilcoxon statistical analysis, we found that the high-risk group had significantly lower IC50 values for Docetaxel, Erlotinib, Gefitinib, Paclitaxel, and Vinorelbine compared to the low-risk group (P value <0.05) (Fig. 7F(2-6)). Cmap scores of molecule drugs in the A549 cell line were shown through the CMap database. Calculate the CMap scores for the five drugs mentioned above based on their IC50 results. The scores of Paclitaxel and Vinorelbine revealed their significant association and more substantial reversal (Table S7). Results suggest that the risk model is sensitive to targeted therapy.

Survival analysis and pathological atlas on single six-HRGs

Kaplan–Meier analysis of six individual HRGs was performed sequentially in GSE72094 to further explore the mechanism of each specific gene (Figs. 8A–8F). Based on the gene expression of six-HRGs, STC1, PFKL, HK1, PDK3, SLC2A1, and XPNPEP1, there was a significant difference in OS between patients in the two risk groups in the training set (log-rank test, P < 0.0001).

Figure 8 The Kaplan–Meier survival analysis and experiments on single six-HRGs.

The Kaplan–Meier survival analysis and experiments on single six-HRGs, namely STC1 (A), SLC2A1 (B), XPNPEP1 (C), PFKL (D), HK1 (E), and PDK3 (F), in GSE72094 dataset to explore the mechanism of each specific gene. (G) Predictive gene expression level validation-PCR. (H) SLC2A1 and XPNPEP1 knockdown efficiency after cell transfection. (I) Effect of knockdown of SLC2A1 on A549 cell proliferation. (J) Effect of knockdown of XPNPEP1 on A549 cell proliferation. (K) Effect of knockdown of SLC2A1 and on A549 cell apoptosis. (L) Effect of knockdown of XPNPEP1 and on A549 cell apoptosis. (M) Effects of knock-down of SLC2A1 and XPNPEP1 on the migration abilities of A549 cells. (N) Effects of knockdown of SLC2A1 and XPNPEP1 on the invasion abilities of A549 cells. A T-test was used to compare the results. * p-value < 0.05, ** p-value < 0.01, *** p-value < 0.001.

Experimental validation on single six-HRGs

Total RNA was extracted from A549 cells and then subjected to RT −PCR experiments. The amplified products were subjected to agarose gel electrophoresis, and six samples all gave clear bands in the correct region indicated by the marker, confirming that the six predicted six-HRGs (STC1, PFKL, HK1, PDK3, SLC2A1, XPNPEP1) were all expressed in lung adenocarcinoma A549 cells (Fig. 8G).

Correlation analysis was performed in TCGA-LUAD dataset using the gene expression levels of six-HRGs. The results, as depicted in Fig. S5, indicated a significant positive correlation between SLC2A1 and XPNPEP1 and the expression levels of other hypoxia-related genes. Two genes listed as risk factors, SLC2A1 and XPNPEP1, were selected for cell transfection and subsequent phenotypic verification experiments. After cell transfection, total RNA was extracted, and the RT–PCR experiment demonstrated that siRNA against both SLC2A1 and XPNPEP1 genes showed knock-down efficiency after transfection (Fig. 8H). Analysis showed that the OD values of A549 cells with exogenous knock-down of the poor prognosis-associated genes SLC2A1 or XPNPEP1 were significantly lower than those of normal control cells after 5 days of culture (P < 0.001), indicating a significant decrease in the proliferation of A549 cells. It reflected a significant reduction in the cellular activity of A549 cells with knock-down of the above two genes (Figs. 8I and 8J). Flow cytometric analysis showed that both SLC2A1 and XPNPEP1 significantly inhibited apoptosis in LUAD cells (P < 0.001, Figs. 8K and 8L). In addition, transwell assays were performed to investigate the effects of predicted poor prognostic genes on LUAD cell migration and invasion. LUAD cells knocked-down with SLC2A1 or XPNPEP1 showed significantly lower migration and invasion rates than cells stimulated with culture medium (P < 0.001, Figs. 8M and 8N). Knock-down of SLC2A1 or XPNPEP1 substantially suppressed the proliferation and migration of A549 cells, demonstrating that the expression of these genes promotes the proliferation of lung adenocarcinoma tumors. This provides a valid reference for the search for biomarkers in the clinical treatment of LUAD.

“Scissor” algorithm to identify subpopulations of TAMs associated with the LUAD hypoxic subtype phenotype in single-cell sequencing data

Using “Scissor”, an algorithm that combines bulk and single-cell sequencing data, the classification of M1 and M2 cells in TAM cells is distinguished in the single-cell data based on the phenotypic differences in the hypoxic subtypes. The validity and guidance value of the identified hypoxic subtypes were assessed. The marker gene identification method was used to differentiate TAMs, where TSPO was used to define M1 macrophages and CD163 was used to define M2 macrophages (Yuan et al., 2022). UMAP shows the distribution of expression of the two marker genes, TSPO and CD163, in TAMs in single-cell data, thus identifying the M1 and M2 cells (Fig. 9A). Scissor analysis of lung cancer single-cell data was performed using the hypoxic subtype in TCGA-LUAD bulk cell samples. Scissor+ cells were predefined as having a hypoxic Cluster 1 presentation profile, and Scissor cells were predefined as having a hypoxic Cluster 2 presentation profile. UMAP plots were viewed for the distribu-tion of Scissor cells, and out of 12,871 cells from different cell types, Scissor se-lected 351 Scissor+ cells and 377 Scissor- cells, which had a high confidence rela-tionship with the M1 and M2 cell classifications in TAMs (Fig. 9B). The analysis showed that the subpopulation of cells closely associated with the hypoxic Cluster 1 phenotype (with a good prognostic effect), was predominantly distributed among M1 cells in single cells, while the subpopulation of cells closely associated with the hypoxic Cluster 2 phenotype (with a poor prognostic effect) was mainly concentrated in M2 cells. Functional enrichment analysis also confirmed that pathways including glycolysis, glucose metabolism, and hypoxia-related pathways were significantly activated in Scissor+ cells (Fig. 9C).

Figure 9 “Scissor” algorithm to identify subpopulations of TAMs associated with the LUAD hypoxic subtype phenotype in single cells.

(A) UMAP diagram showing the distribution of TSPO, CD163 expression in TAMs of scRNA-seq of LUAD. (B) Scissor for visualization of UMAP of selected cells. The red and blue dots are LUAD hypoxic Cluster 1 and LUAD hypoxic Cluster 2. (C) Selected enrichment bars associated with hypoxia.

Discussion

This study integrated bioinformatics analysis and experimental validation to investigate the impact of hypoxia-related genes on immune mechanisms and clinical prognosis in lung adenocarcinoma. We used clustering methods to identify hypoxia-related subtypes and demonstrated significant differences in the immune microenvironment and immune cell composition among these subtypes using GSVA and ssGSEA methods. Subsequently, we constructed a prognostic model composed of 6 hypoxia-related genes using Cox regression and Lasso regression. The prognostic model effectively defined TAMs cell subgroups in the single-cell queue of lung adenocarcinoma and showed differences in enriched pathways and immune therapies between the risk groups. To further evaluate the impact of hypoxia-related genes on lung adenocarcinoma cells, we conducted RT-PCR and in vitro experiments, focusing on the expression of SLC2A1 and XPNPEP1 genes which promote the proliferation of lung adenocarcinoma cells. This study provides potential hypoxia-related prognostic markers for clinical treatment of lung adenocarcinoma and supports clinical researchers in conducting prognostic treatment.

The treatment effects of LUAD remain suboptimal, with a current 5-year OS rate of only 15% for patients with LUAD (Guo et al., 2020). Studies have shown that tumor cells are in a specific microenvironment. On the one hand, tumor cells need to consume a large amount of oxygen due to rapid growth and proliferation (Jing et al., 2019); on the other hand, the corresponding vascular system is slow to develop and less oxygen is available due to the overgrowth of tumor tissue (Farina et al., 2020). Hypoxia has been found to be a possible driver of tumor metastasis and to promote the invasive capacity of lung cancer cells (Liu et al., 2018b). Hypoxia can induce tumor cells to express various genes to adapt to the hypoxic microenvironment. This has inspired researchers to analyze how relevant hypoxia genes exert potential biofunctional mechanisms to influence tumor cell growth migration in clinical patients with LUAD, providing a reference for patient survival prediction.

In this study, we used a consistent clustering approach to delineate subgroups in the four LUAD cohorts based on prognosis-related hypoxia genes, with all groupings resulting in K = 2. Prognostic analysis in four cohorts all showed significant survival differences between the hypoxic subgroups, with patients in the hypoxic Cluster 1 subgroup having significantly better survival than Cluster 2. Further immune analysis and clinical characterization between the hypoxic subgroups were performed, showing a significant expression of immune-related pathways in the hypoxic Cluster 1 subgroup, which revealed a significant correlation between hypoxia and immune regulatory mechanisms in TME, resulting in significant differences in clinical phenotypes. Subsequently, we used the “Scissor” algorithm to analyze the role of phenotypic differences in hypoxic subpopulations identified on single cells by integrat-ing bulk and scRNA-seq data. M1 cells have been reported to have anticancer, proinflammatory, and antitumor effects, whereas M2 cells have procancer effects and are involved in immune regulation and repair functions (Nakagawa & Chiba, 2015). Therefore, M1 and M2 cell subsets in LUAD monocytes were selected as a single-cell data source, and the results showed that the hypoxic Cluster 1 subtype with good prognosis was mainly distributed in M1 cells, and the hypoxic Cluster 2 subtype with poor prognosis was mainly in M2 cells. The value and validity of hypoxic subtype grouping were verified by single-cell analysis. Both bulk and single-cell sequencing data demonstrate significant differences between the delineated hypoxic subtypes, resulting in differences in immune infiltration characteristics in the TME due to differential expression of hypoxic genes and causing different pathway en-richment outcomes, ultimately presenting different clinical phenotypic differences and prognostic outcomes. The application of the “scissor” algorithm allowed the study to explore both bulk transcriptomic and single-cell data, enhancing the diversity of data types, increasing the credibility of the study, and demonstrating that the hypoxic subtypes classified based on cluster analysis were validated in both bulk and single-cell data, which is a good reference for clinical treatment.

Through data mining work on canceromics databases, we constructed patient cohorts containing one training dataset and three validation datasets from the GEO and TCGA databases. Using classical machine learning analysis, a prognostic model was constructed. After the training and optimization of the model, six HRGs were identi-fied as prognostic markers. A risk score consisting of the six HRG features could predict the survival outcome of patients. We next explored the impact of six HRGs on lung adenocarcinoma in detail. Single-gene prognostic analysis of the six HRGs with prognostic signatures was performed, and all 6 genes could significantly distinguish between the two risk groups. The paired-sample T test and PCR experiments further illustrated that each of the six genes was significantly expressed in LUAD cancer cells and significantly higher than in normal tissues of clinical patients.

Several studies have shown that solute carrier family 2 member 1 (SLC2A1) is overexpressed in the tumor tissues of LUAD patients and may be involved in tumor-igenesis (Cheng et al., 2021). SLC2A1 can be used as a significant risk factor affecting the survival of LUAD patients (Liu et al., 2018a). STC1 is a secreted growth factor-like glycoprotein that binds to membrane-bound receptors and acts in an autocrine and paracrine manner  (Song et al., 2022). As a hypoxia gene, STC1 is also an intracellular protein that can bind to mitochondrial receptors, increase mitochondrial oxidative phosphorylation (OXPHOS), and play an important role in epithelial-mesenchymal transition (EMT) and cancer stem cell (CSC) formation (Ellard et al., 2007; Zhang et al., 2000; Abe et al., 2021). STC1 can induce the adaptive response of human cancer cells to hypoxia by regulating hypoxia-inducible factor-1-α (HIF-1α), and is abnormally expressed in a variety of cancer types (Cao et al., 2019). In lung cancer, overexpression of PFKL (phosphofructokinase) has been found to promote cell growth, colony formation, and decrease cellular ATP content, while also predicting poor overall survival (Yang et al., 2016). PFKL is associated with the predicting poor overall lung cancer patient survival. Hypoxic conditions upregulate HK1 (hexokinase 1), a critical glycolytic enzyme (Cheng et al., 2018), which has been linked to poor prognosis in malignant melanoma, head and neck squamous cell carcinoma, and oral squamous cell carcinoma (Zhou et al., 2019; Han et al., 2021). PDK3 (pyruvate dehydrogenase kinase 3) is another enzyme that promotes glycolysis in tumor cells (Lu et al., 2008). and its high expression contributes to drug resistance induced by hypoxia in most colon cancer cells, making it a potential target for cancer treatment (Ren et al., 2017). PNPEP1 (aminopeptidase P) is specifically expressed in lung tumors and presented in pulmonary blood vessels, and high expression of XPNPEP1 in multiple myeloma plasma cells has been associated with decreased overall survival (Miettinen et al., 2021). STC1 is an oncogene that regulates cellular processes in normal development and tumorigenesis (Chang et al., 2015; Sun et al., 2021; Tse et al., 2020). The expression of six hypoxia-related genes are all closely related to the survival outcomes of patients with various types of cancer.

Therefore, additional experimental analyses were used to test the effect of hypoxia genes in LUAD cell lines after validation based on model training and literature support. Based on correlation analysis, we selected two prognostic genes, SLC2A1 and XPNPEP1, which according to our prediction model are poor prognostic genes. Experimental analyses were performed in A549 cells, including cell transfection analysis, cell proliferation analysis, apop-tosis assay, and invasion assay. The results showed that the poor phenotype of tumor cells was significantly improved after knock-down of these two genes. The experi-ments demonstrated that the six genes in the prediction model were indeed expressed in lung adenocarcinoma cells, and the correctness of the prediction model were confirmed by observing the changes in cell phenotype after the genes associated with poor prognosis were knocked-down and treated.

As a biomarker, TMB can reflect the number of cancer mutations (Barroso-Sousa et al., 2020). Many studies have focused on analyzing the potential impact of TMB levels on ICI treatment. Alterations in the metabolic and immune aspects of the TME environment can limit and influence tumorigenesis. Tumor mutations cause the production of immunogenic neoantigens on the surface of tumor cells, facilitating their recognition during immunotherapy. A higher TMB index indicates a higher number of tumor mutations and a higher likelihood that certain neoantigens presented by MHC proteins are immunogenic. Thus, high levels of TMB are associated with better ICI treatment outcomes (Lu et al., 2019). We performed a correlation analysis between TMB scores and risk scores constructed based on 6-HRGs. The results showed that the more significant the TMB mutation, the higher the risk score. Combining the LUAD prognostic model with the TMB mutation score is a potential guide to the clinical treatment of ICI for cancer.

The immune infiltration of tumor cells has been associated with LUAD prognosis. The MCP-counter algorithm was used to investigate this relationship, revealing that CAF expression levels were significantly higher in the high-risk group, while B cell and T cell expression levels were higher in the low-risk group. These findings suggest that the immune response to the tumor microenvironment of LUAD, particularly involving B cells, T cells, and CAF, may be influenced by risk groups. B cells and T cells are crucial immune cells that possess broad anti-tumor effects (Li et al., 2021; Fiori, Villanova & De Maria, 2017; Germain et al., 2014). Germain (Yang et al., 2013) reported that a high density of B cells was associated with a better prognosis in lung cancer patients. LUAD patients in the B-cell low expression group have a poorer prognosis (Liu et al., 2021). FoxP3 is a crucial transcription regulator factor for regulatory T cells (Tregs) (Ben-Shoshan et al., 2008). Clambey et al. (2012) demonstrated through in vitro and mouse experiments that under long-term hypoxia conditions, HIF-1α promotes FoxP3 expression, enhancing Treg abundance and triggering a potent anti-inflammatory mechanism that limits tissue damage under insufficient oxygen supply (Shi et al., 2011). Immune infiltration analysis showed significantly decreased T cell expression in the high-risk group, which affects the body’s ability to exert anti-inflammatory effects, suppress cancer cell proliferation, and ultimately leads to a significant decrease in patient survival rate.

The tumor microenvironment is a complex ecosystem that includes CAFs, immune cells, etc. CAFs promote tumor vascular growth and enhance the inhibition of apoptosis in the TME (Liu et al., 2019). Hypoxia is also an essential component of the TME. Solid tumor tissues are usually hypoxic and have a limited number of fully functional blood vessels (Hockel & Vaupel, 2001; Semenza, 2012). The hypoxic TME affects many biological activities, such as metabolism, and cell proliferation, and these changes can promote tumor development and inhibit tumor therapy. All these studies revealed the role of hypoxia in CAF activation and its role in the TME. According to the results of the immune infiltration analysis, CAF expression was significantly higher in the high-risk group. CAFs would survive better in the TME with very high expression of the HRG signature. This suggests that CAFs in lung adenocarcinoma cells have a similar survival status to other cancer species under hypoxic conditions. Prolonged exposure to the TME under hypoxic conditions is a complex network of immune regulation that ultimately promotes tumor growth (Feng et al., 2022).

GSEA was performed to explore the molecular mechanisms by which 6-HRG intervenes in lung adenocarcinogenesis and progression. Glycosylphosphatidylino-sitol (GPI)-anchor biosynthesis is the pathway that exhibits significant activation in people at high risk for LUAD. Due to hypoxia and a low pH microenvironment, tu-mor cells exhibit high glycolysis, known as the Warburg effect, rather than oxidative phosphorylation in normal cells (Cairns, Harris & Mak, 2011; Doherty & Cleveland, 2013). Through multiple signaling pathways, GPI-anchored cell surface proteins inhibit the immune response and promote tumor immune escape in tumor cells in vivo (Hanahan & Weinberg, 2011). In LUAD, prolonged hypoxic conditions shape the high expression of relevant hypoxic genes. Hypoxic genes activate GPI-anchor biosynthesis pathway, leading to abnormal intracellular energy metabolic activities that can promote related oncogenic phenotypes such as rapid cell proliferation and drug resistance (Gatenby & Gillies, 2004). In MPNST patients, SLC2A1 provides glucose influx to maintain anaerobic glycolysis, which ultimately contributes to the “Warburg” effect  (Krawczyk et al., 2021). This study further showed that SLC2A1 can promote tumor cell growth by activating glycolysis-related pathways in different cancer species.

Combined with the results of IC50 and Cmap analysis, paclitaxel and vinorelbine had significantly lower IC50 values and their Cmap scores were below −0.7, suggesting that the drug candidates have a good reversal mechanism for LUAD disease. Paclitaxel and vinorelbine are widely available (Lu et al., 2008; Lu et al., 2011) because the FDA has licensed them for the treatment of NSCLC. Our study suggests that they are not only therapeutic for patients in the LUAD cohort but also have a better potential therapeutic effect in high-risk populations. Notably, the analysis found that patients in the high-risk group had a worse prognosis but were more likely to benefit from ICB and drug therapy.

In order to comprehensively assess the influence of hypoxia-related genes on LUAD, this study integrated bioinformatics analysis with cellular experiments. Our future research plan involves incorporating animal experiments, such as mice, to further investigate the role of critical hypoxia genes in modulating the proliferation and immune infiltration levels of LUAD tumors.

Conclusions

In summary, our study is based on an unsupervised clustering algorithm to identify hypoxic subtypes. Differences in the TME and immunotherapy between the two hypoxic subtypes were compared. A six-HRG signature that could predict the prognosis of LUAD patients was constructed, facilitating the prediction of survival outcomes and potential treatment options for LUAD patients.

Supplemental Information

Figure S1 Consensus Clustering and Kaplan-Meier survival curves for the hypoxic subtypes Cluster 1 and Cluster 2 in the GSE72094 cohort (A, B), GSE31210 cohort (C, D), and GSE30219 cohort (E, F)

Click here for additional data file.

Figure S2 Risk score between subtypes of Age (A), Gender (B), Pathological stage (C), Pathological T (D), Pathological N (E) and Pathological M (F) in TCGA-LUAD cohort

* p-value < 0.05, ** p-value < 0.01, *** p-value < 0.001.

Click here for additional data file.

Figure S3 The prognostic ability of the six-HRG signature for overall survival in multiple LUAD subtypes in TCGA-LUAD cohort

Kaplan-Meier curves for OS prediction in LUAD subtypes of (A) Age <70 years, (B) Age ≥70 years, (C) Male, (D) Female, (E) Pathological stage I-II, (F) Pathological stage III-IV, (G) Pathological T1-2, (H) Pathological T3-4, (I) Pathological N0, (J) Pathological N1-3, (K) Pathological M0, (L) Pathological M+.

Click here for additional data file.

Figure S4 Estimates of tumor-infiltrating immune cells were calculated separately for the risk groups in GEO cohort

GSE72094 (A), GSE31210 (B) and GSE30219 (C) cohorts.

Click here for additional data file.

Figure S5 Correlation analysis among the 6 hypoxia-related genes

Click here for additional data file.

Supplemental Information 6 Supplemental Tables

Click here for additional data file.

Data S1 Raw datas and codes

Raw data including raw GEO and TCGA processing data, as well as relevant processing codes for prognostic analysis, and immunoassays.

Click here for additional data file.

Supplemental Information 8 RT-PCR experiment results (uncropped)

Click here for additional data file.

Additional Information and Declarations

Competing Interests

Author Contributions

Data Availability

The authors declare there are no competing interests.

Liu Liu conceived and designed the experiments, performed the experiments, analyzed the data, prepared figures and/or tables, and approved the final draft.

Lina Han conceived and designed the experiments, performed the experiments, prepared figures and/or tables, and approved the final draft.

Lei Dong conceived and designed the experiments, performed the experiments, prepared figures and/or tables, and approved the final draft.

Zihao He analyzed the data, prepared figures and/or tables, and approved the final draft.

Kai Gao analyzed the data, prepared figures and/or tables, and approved the final draft.

Xu Chen performed the experiments, prepared figures and/or tables, and approved the final draft.

Jin-Cheng Guo conceived and designed the experiments, authored or reviewed drafts of the article, and approved the final draft.

Yi Zhao conceived and designed the experiments, authored or reviewed drafts of the article, and approved the final draft.

The following information was supplied regarding data availability:

The raw measurements are available in the Supplementary Files.

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
