# Peer review of "The hypoxia-associated genes in immune infiltration and treatment options of lung adenocarcinoma"

_PeerJ, doi:10.7717/peerj.15621_

## Round 0.1 · original submission · Major Revisions

Dear Dr. Liu,

Please address all the reviewer comments.

Reviewer 1 ·

Basic reporting

The English language should be improved. There are grammar mistakes and some sentences are confusing. The logic manner should also be improved. The introduction should be further refined to emphasize the novelty and clinical value of the study. The article structure is ok. Raw data shared.

Experimental design

1. This is a study based on bioinformatics analysis. The article uses a variety of scoring systems, such as TMB, immunoscore. Based on the results of omics data analysis and hypoxia-related genes, a clinical scoring system is suggested to be summarized with consideration of hypoxia-related genes and related immune indicators.

2. In order to further verify the clinical application value of the prognostic features of hypoxia-related genes and their impact on the tumor microenvironment of lung adenocarcinoma, clinically relevant specimens can be collected to analyze the expression of hypoxia-related genes, verify the results of big data analysis, and these Correlation analysis between genes and clinical factors such as advanced TNM stage, pathological stage, histological grade, and immune infiltration.

Validity of the findings

The correlation between the expression of hypoxia-related genes in tissue samples and the distribution of immune cells in the tumor microenvironment can be further verified. The effects of hypoxia genes on the tumor immune microenvironment of lung adenocarcinoma can be verified in various ways by immunohistochemistry or flow cytometry. If necessary, further in vitro cell or animal in vivo experiments can be carried out.

Reviewer 2 ·

Basic reporting

Dear Author,
Below are my queries and suggestions about your article titled " The hypoxia-associated genes in immune infiltration and treatment options of Lung Adenocarcinoma".
General issues:
1. The paper needs a grammar check.
2. Replace was with were in line 27, Study with study in line 43, and was with were in line 447.
3. You have written the same word with and without a hyphen in your document as Knock-down or knockdown in manuscript and figure legends. Also, the word Signature or signature. It's best to be consistent.

Experimental design

Materials and Methods:
1. In line 99, replace the text (further screen) with (screen further).
2. Uniform all figures and tables in brackets.

Validity of the findings

Results:
1. In line 299, replace the text (further assess) with (assess further).
2. Uniform all figures and tables in brackets.
3. In line 369, replace 10 with Ten.
4. In line 377, Uniform the word Six HRGs or six as in line 379.

Additional comments

Discussion:
Small info should be given before starting the discussion.
References: Overall OK.

Reviewer 3 ·

Basic reporting

In the presented manuscript, Liu et. al., aimed to show hypoxia-associated genes and immune cell infiltration during Lung adenocarcinoma using RNA -seq data analysis. The authors expanded their findings by validating observed genes for their expression and function in lung cancer cell line A549. This study also highlights important immune cells and TAMs association in hypoxic groups with tumor prognosis. However, the data overall is based on RNA seq and little experimental data.

Experimental design

-Results section 7, Authors describing immune cell infiltration in the high and low-risk groups. The explanation in results lacks any reference to literature for any observed cell types. Please add references for all described observations for scientific rigor.
- Figure 7A, the differences in Immune cells Lmmune (probably it is ‘Immune’) score is tiny to be mentioned for the significant outcome, especially, T cells. These data points are highly variable to signify their respective differences. If authors can justify this with available experimental literature this data makes more sense.
- Similarly, the TIDE score is also very less different in the two groups and does not show a significant change (7F). Please add more explanations for the observations for better understanding.
-Fig. 7H, the IC50 score is not big enough to claim a difference between the two groups. If so, please add correct references showing a similar level of difference with significant effect.
-In results section 8 (Figure 8), the authors have shown the expression of 6 predicted HRGs in A549 cells. However, results lack an explanation of these genes and their association with tumor progression and prognosis. The author must include information about each gene with appropriate literature references.
-in the result section 8, the authors selected two genes for further validation experiments. This section should include the rationale for this preference for analysis.
- Panel (8I, 8J), results description directly used OD value for cellular activity measurement without explaining the specifics for experiment reasons and details. What is OD denoting here and why it has shown?
-In the conclusions, the authors wrote “Data suggested that high expression of SLC2A1 and 361
XPNPEP1 significantly promotes cell proliferation in LUAD, leading to poor prognosis”.
Authors should correct and rephrase the conclusions on what direct data show (knockdown of these genes, reduced apoptosis, proliferation, and migration) and how it can be interpreted for high expression conditions.

Validity of the findings

The observed difference in critical tumor factors, e.g. immune cells, and genes show minuscule differences in scores. Therefore, the validity of id data hardly satisfies the claims.

---

## Round 0.2 · accepted · Accept

Dear Dr. Liu,
Thank you for your submission has been Accepted for publication.

Reviewer 1 ·

Basic reporting

My comments have been addressed satisfactorily.

Experimental design

My comments have been addressed satisfactorily.

Validity of the findings

My comments have been addressed satisfactorily.

Reviewer 2 ·

Basic reporting

The author has made all the suggested modifications.

Experimental design

The author has made all the suggested modifications.

Validity of the findings

The author has made all the suggested modifications.

Reviewer 3 ·

Basic reporting

The author has included all the corrections raised by me.

Experimental design

Current form of the manuscript is in a publishable state.

Validity of the findings

Due to added data, manuscript findings support the results.